# Spatial modulation of visual responses arises in cortex with active navigation

E Mika Diamanti[1,2]*, Charu Bai Reddy[1], Sylvia Schröder[1], Tomaso Muzzu[3], Kenneth D Harris[4], Aman B Saleem[3]*, Matteo Carandini[1]

[1]UCL Institute of Ophthalmology, University College London, London, United Kingdom; [2]CoMPLEX, Department of Computer Science, University College London, London, United Kingdom; [3]UCL Institute of Behavioural Neuroscience, University College London, London, United Kingdom; [4]UCL Queen Square Institute of Neurology, University College London, London, United Kingdom

**Abstract** During navigation, the visual responses of neurons in mouse primary visual cortex (V1) are modulated by the animal's spatial position. Here we show that this spatial modulation is similarly present across multiple higher visual areas but negligible in the main thalamic pathway into V1. Similar to hippocampus, spatial modulation in visual cortex strengthens with experience and with active behavior. Active navigation in a familiar environment, therefore, enhances the spatial modulation of visual signals starting in the cortex.

## Introduction

There is increasing evidence that the activity of the mouse primary visual cortex (V1) is influenced by navigational signals (*Fiser et al., 2016*; *Flossmann and Rochefort, 2021*; *Fournier et al., 2020*; *Haggerty and Ji, 2015*; *Ji and Wilson, 2007*; *Pakan et al., 2018*; *Saleem et al., 2018*). During navigation, indeed, the visual responses of V1 neurons are modulated by the animal's estimate of spatial position (*Saleem et al., 2018*). The underlying spatial signals covary with those in hippocampus and are affected similarly by idiothetic cues (*Fournier et al., 2020*; *Saleem et al., 2018*).

It is not known, however, how this spatial modulation varies along the visual pathway. Spatial signals might enter the visual pathway upstream of V1, in the lateral geniculate nucleus (LGN). Indeed, spatial signals have been seen elsewhere in thalamus (*Jankowski et al., 2015*; *Taube, 1995*) and possibly also in LGN itself (*Hok et al., 2018*). Spatial signals might also become stronger downstream of V1, in higher visual areas (HVAs). For instance, they might be stronger in parietal areas such as A and AM (*Hovde et al., 2018*), because many neurons in parietal cortex are associated with spatial coding (*Krumin et al., 2018*; *McNaughton et al., 1994*; *Nitz, 2006*; *Save and Poucet, 2009*; *Whitlock et al., 2012*; *Wilber et al., 2014*).

Moreover, it is not known if the spatial modulation of visual responses varies with experience in the environment or active navigation. In the navigational system, spatial encoding is stronger in active navigation than during passive viewing, when most hippocampal place cells lose their place fields (*Chen et al., 2013*; *Song et al., 2005*; *Terrazas et al., 2005*). In addition, both hippocampal place fields and entorhinal grid patterns grow stronger when an environment becomes familiar (*Barry et al., 2012*; *Frank et al., 2004*; *Karlsson and Frank, 2008*). If spatial modulation signals reach visual cortex from the navigational system, therefore, they should grow with active navigation and with experience of the environment.

## Results

To characterize the influence of spatial position on visual responses, we used a virtual reality (VR) corridor with two visually matching segments (*Saleem et al., 2018*; *Figure 1*). We used two-photon

*For correspondence:
emdiamanti@princeton.edu (EMD);
aman.saleem@ucl.ac.uk (ABS)

Competing interests: The authors declare that no competing interests exist.

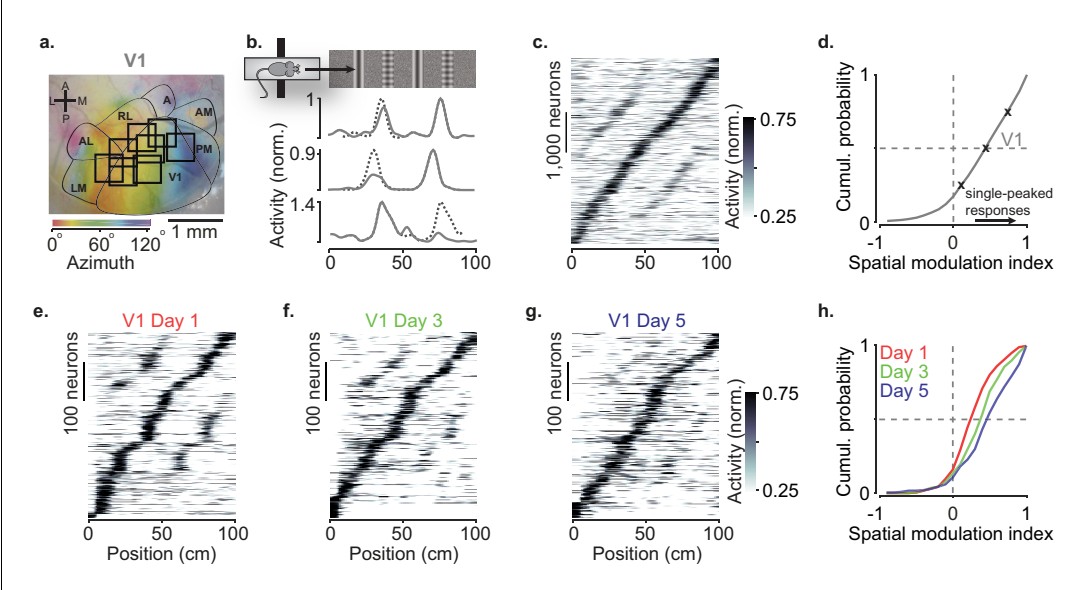

**Figure 1.** Spatial modulation strengthens with experience. (**a**) Example retinotopic map (*colors*) showing borders between visual areas (*contours*) and imaging sessions targeting V1 fully or partly (*squares*, field of view: 500 × 500 µm). (**b**) Normalized responses of three example V1 neurons, as a function of position in the virtual corridor. The corridor had two landmarks that repeated after 40 cm, creating visually matching segments (*top*). *Dotted lines* are predictions assuming identical responses in the two segments. (**c**) Responses of 4602 V1 neurons (out of 16,238) whose activity was modulated along the corridor (≥5% explained variance), ordered by the position of their peak response. The ordering was based on separate data (odd-numbered trials). (**d**) Cumulative distribution of the spatial modulation index (SMI) for the V1 neurons. Only neurons responding within the visually matching segments are included (2992/4602). *Crosses* mark the 25th, 50th, and 75th percentiles and indicate the three example cells in (**b**). (**e–g**) Response profiles obtained from the same field of view in V1 across the first days of experience of the virtual corridor (days 1, 3, and 5 are shown) in two mice. (**h**) Cumulative distribution of SMI for those 3 days, showing median SMI growing from 0.24 to 0.38 to 0.45 across days.
The online version of this article includes the following figure supplement(s) for figure 1:

**Figure supplement 1.** Lateral geniculate nucleus (LGN) boutons tile up the virtual corridor differently from V1 neurons.

microscopy to record activity across the visual pathway, from neurons in layer 2/3 of multiple visual areas, and from LGN afferents in layer 4 of area V1. To estimate activity, the calcium traces were deconvolved, yielding inferred firing rates (*Pachitariu et al., 2016a*; *Pachitariu et al., 2018*). Mice were head-fixed and ran on a treadmill to traverse a one-dimensional virtual corridor made of two visually matching 40 cm segments each containing the same two visual patterns (*Figure 1b*, *top*). A purely visual neuron would respond to visual patterns similarly in both segments, while a neuron modulated by spatial position could respond more strongly in one segment.

As we previously reported, spatial position powerfully modulated the visual responses of V1 neurons (*Figure 1a–d*). V1 neurons tended to respond to the visual patterns more strongly at a single location (*Fournier et al., 2020*; *Saleem et al., 2018*; *Figure 1b*) and their preferred locations were broadly distributed along the corridor (*Figure 1c*, *Figure 1—figure supplement 1a–c*). To quantify this spatial modulation of visual responses, we defined a spatial modulation index (SMI) as the normalized difference of responses at the two visually matching positions (preferred minus non–preferred, divided by their sum, with the preferred position defined on held-out data). The distribution of SMIs across V1 neurons heavily skewed toward positive values (*Figure 1d*), which correspond to a single peak as a function of spatial position. The median SMI for responsive V1 neurons was 0.39 ± 0.19 (n = 39 sessions) and 44% of V1 neurons (1322/2992) had SMI > 0.5.

Spatial modulation in V1 grew with experience (*Figure 1e–h*). In two mice, we measured spatial modulation across the first 5 days of exposure to the virtual corridor, imaging the same V1 field of view across days. Response profiles on day 1 showed many responses with two pronounced peaks (*Figure 1e*). By day 5, response profiles were more single-peaked and resembled those recorded in experienced mice (*Figure 1c,f,g*). Indeed, the spatial modulation increased with experience and was significantly larger on day 5 compared to day 1 (*Figure 1h*, median SMI: 0.45 on day 5 vs. 0.24 on day 1; $p < 10^{-12}$, two-sided Wilcoxon rank sum test).

In contrast to V1 neurons, spatial position barely affected the visual responses of LGN afferents in experienced mice (*Figure 2a–d*). LGN boutons in layer 4 gave mostly similar visual responses in the two segments of the corridor (*Figure 2b,c*) and the locations where they fired clustered around the landmarks as expected from purely visual responses (*Figure 1—figure supplement 1e*). The spatial modulation in LGN boutons was small (*Figure 2d*), with a median SMI barely above zero (median ± m.a.d.: 0.07 ± 0.05, n = 19 sessions). It was slightly positive (p=0.002, right-tailed Wilcoxon signed rank test), but markedly smaller than the SMIs of V1 neurons (p=$10^{-6}$, left-tailed Wilcoxon rank sum test). Only 4% of LGN boutons (37/840) had SMI > 0.5, compared to 44% in V1. Moreover, among these few neurons, most (28/37) fired more strongly in the first half of the corridor (*Figure 1—figure supplement 1f*), as might be expected from contrast adaptation mechanisms (*Dhruv and Carandini, 2014*).

Similar results were seen in recordings from LGN neurons (*Figure 2—figure supplement 1*). We performed extracellular electrophysiology recordings in LGN (two mice, five sessions). LGN units gave responses and SMI similar to LGN boutons (median ± m.a.d.: 0.06 ± 0.02, p=0.78, Wilcoxon rank sum test). Median SMI was slightly positive (p=0.03, right-tailed Wilcoxon signed rank test), but again markedly smaller than in V1 (p=0.002, left-tailed Wilcoxon rank sum test).

Spatial modulation was broadly similar across HVAs and not significantly larger than in V1 (*Figure 2e–h*, *Figure 3*). We measured activity in six visual areas that surround V1 (LM, AL, RL, A, AM, and PM) and found strong modulation by spatial position (*Figure 2f,g*). Pooling across these areas, the median SMI across sessions was 0.40 ± 0.12, significantly larger than zero (n = 52 sessions, p=$10^{-10}$, right-tailed Wilcoxon signed rank test, *Figure 2h*) and not significantly different from V1 (Wilcoxon rank sum test: p=0.88). Spatial modulation was present in each of the six areas (*Figure 3*) and, as in V1, was not affected by reward protocol or mouse line (*Figure 3—figure supplement 3*). In addition, spatial modulation could not be explained by other factors such as running speed, reward events, pupil size, and eye position (*Figure 3—figure supplement 1*).

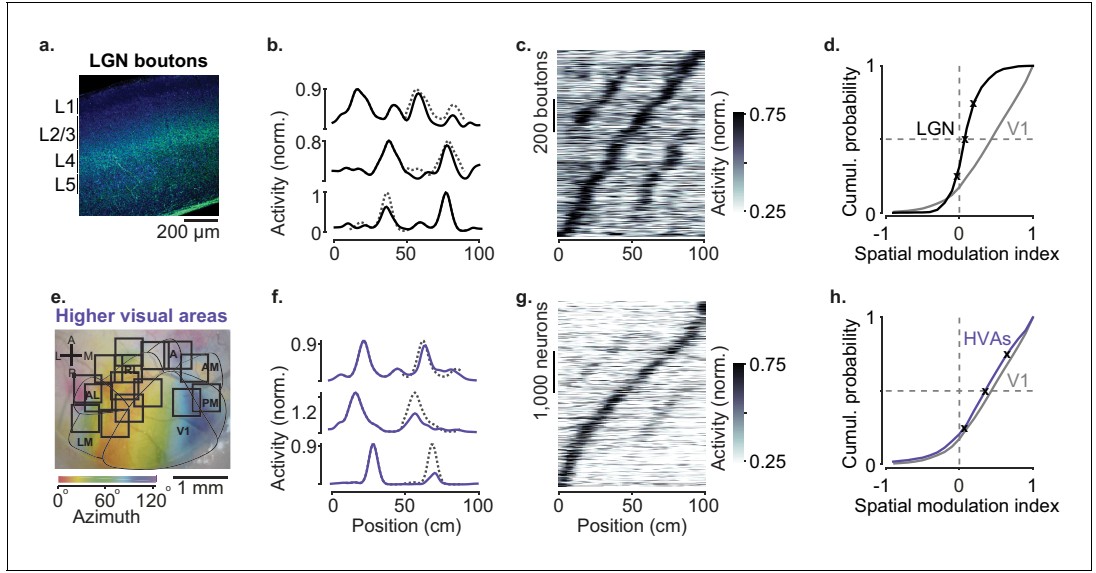

**Figure 2.** Modulation of visual responses along the visual pathway during navigation. (a) Confocal image of lateral geniculate nucleus (LGN) boutons expressing GCaMP (GFP; *green*) among V1 neurons (Nissl stain; *blue*). GCaMP expression is densest in layer 4 (L4). (b) Normalized activity of three example LGN boutons, as a function of position in the virtual corridor. *Dotted lines* are predictions assuming identical responses in the two segments. (c) Activities of 1140 LGN boutons (out of 3182) whose activity was modulated along the corridor (≥5% explained variance), ordered by the position of their peak response. The ordering was based on separate data (odd-numbered trials). (d). Cumulative distribution of the spatial modulation index (SMI) for the LGN boutons. Only boutons responding within the visually matching segments are included (LGN: 840/1140). *Crosses* mark the 25th, 50th, and 75th percentiles and indicate the three example cells in (b). (e) Same as in *Figure 1a*, showing imaging sessions targeting six higher visual areas (HVA) fully or partly. (f–h) Same as (b–d), showing response profiles of HVA neurons ((g) 4381 of 18,142 HVA neurons; (h) 2453 of those neurons).

The online version of this article includes the following figure supplement(s) for figure 2:

**Figure supplement 1.** Lateral geniculate nucleus (LGN) boutons and units give similar responses.

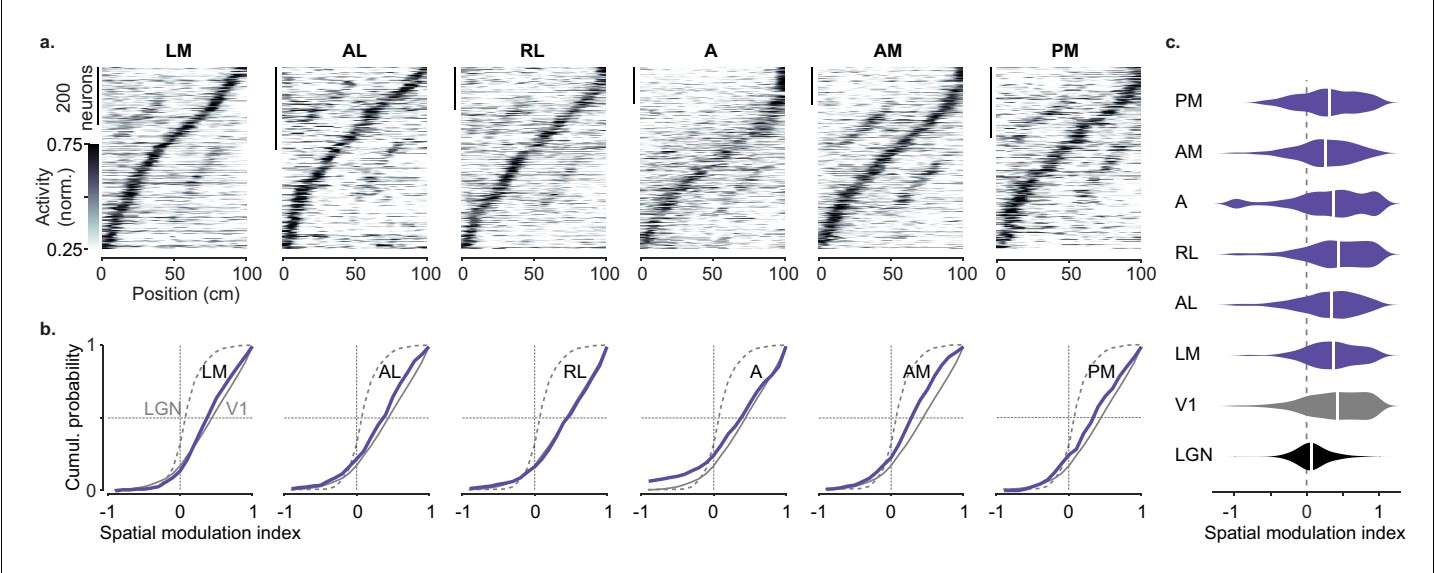

**Figure 3.** Spatial modulation of individual higher visual areas (HVAs). (a) Response profile patterns obtained from even trials (ordered and normalized based on odd trials) for six visual areas. Only response profiles with variance explained ≥5% are included (LM: 629/1503 AL: 443/1774 RL: 866/5192 A: 997/4126 AM: 982/3278 PM: 519/2509). (b). Cumulative distribution of the spatial modulation index in even trials for each HVA (*purple*). *Dotted line*: lateral geniculate nucleus (LGN; same as in *Figure 2d*), *Gray*: V1 (same as in *Figure 1d*). (c). Violin plots showing the spatial modulation index (SMI) distribution and median SMI (*white vertical line*) for each visual area (median ± m.a.d. LGN: 0.07 ± 0.11; V1: 0.43 ± 0.31; LM: 0.37 ± 0.25; AL: 0.34 ± 0.28; RL: 0.44 ± 0.31; A: 0.37 ± 0.34; AM: 0.27 ± 0.26; PM: 0.32 ± 0.32).

The online version of this article includes the following figure supplement(s) for figure 3:

**Figure supplement 1.** Spatial modulation is not explained by other behavioral and visual factors.

**Figure supplement 2.** Neurons with central receptive fields showed stronger spatial modulation than neurons with peripheral receptive fields due to the layout of the visual scenes.

**Figure supplement 3.** Spatial modulation does not depend on reward or mouse line.

We observed small differences in spatial modulation between areas, which may arise from biases in retinotopy combined with the layout of the visual scene. Visual patterns in the central visual field were further away in the corridor, and thus were likely less effective in driving responses than patterns in the periphery, which were closer to the animal and thus larger. In V1, spatial modulation was larger in neurons with central rather than peripheral receptive fields (*Figure 3—figure supplement 2*) irrespective of mouse line or reward protocol (*Figure 3—figure supplement 3*). A similar trend was seen across higher areas, with slightly lower SMI in areas biased toward the periphery (AM, PM) (*Garrett et al., 2014*; *Wang and Burkhalter, 2007*; *Zhuang et al., 2017*) than in areas biased toward the central visual field (LM, RL, *Figure 3c*).

We next asked whether visual responses would be similarly modulated when animals passively viewed the environment. After recordings in 'VR', we played back the same video regardless of the mouse's movements ('replay'). We separated data taken during running (running speed >1 cm/s in at least 10 trials, 'running replay'), and rest ('stationary replay') periods.

Passive viewing affected the baseline activity of LGN boutons but not their spatial modulation, which remained negligible in all conditions (*Figure 4a–d*). During 'running replay' the baseline activity of LGN boutons decreased slightly (*Figure 4a,b*, *Figure 4—figure supplement 1a*, p=0.003, paired-sample right-tailed t-test). However, the median SMI in 'running replay' remained a mere 0.05 ± 0.06 (n = 18 sessions), not significantly different from the 0.08 ± 0.05 measured in VR (*Figure 4b*, p=0.29, Wilcoxon signed rank test). Similar results were obtained during stationary replay (*Figure 4c,d*): baseline activity decreased markedly (*Erisken et al., 2014*) (p=$10^{-65}$ paired-sample right-tailed t-test, *Figure 4—figure supplement 1b*), but the median SMI remained negligible at 0.03 ± 0.04 and not different from the 0.07 ± 0.04 measured in VR (n = 18 sessions, p=0.053, Wilcoxon signed rank test, *Figure 4d*).

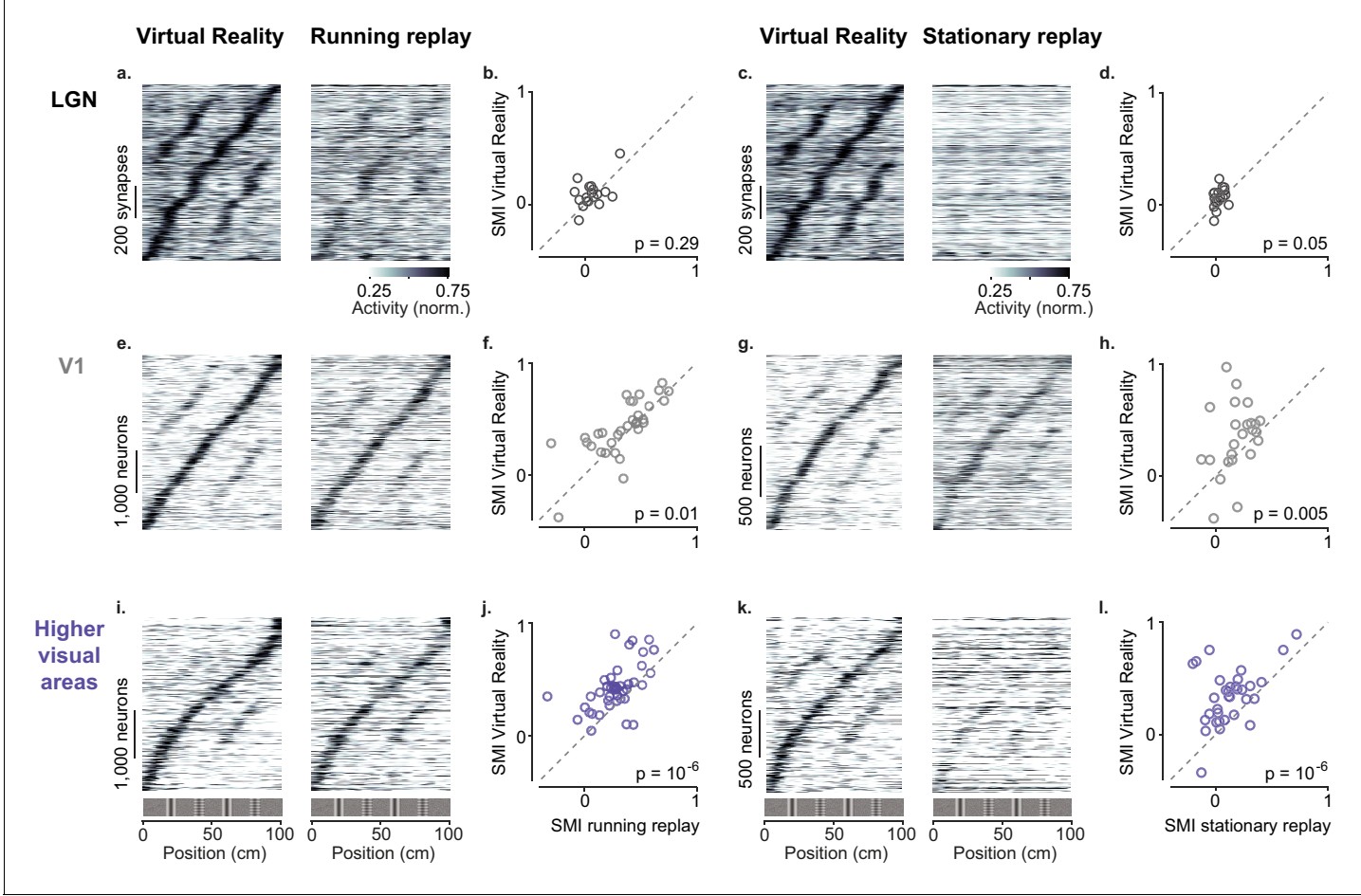

**Figure 4.** Active navigation enhances modulation by spatial position in visual cortical areas. (a) Response profiles of lateral geniculate nucleus (LGN) boutons in virtual reality (VR; *left*) that also met the conditions for running replay (*right*; at least 10 running trials per recording session), estimated as in *Figure 1c*. Response profiles of LGN boutons during running replay were ordered by the position of their maximum response estimated from odd trials in VR (same order and normalization as in *left* panel). (b) Median spatial modulation index (SMI) per recording session in VR versus running replay for LGN (each circle corresponds to a single session; p-values from Wilcoxon signed rank test). (c, d) Same as (a, b) for stationary replay. (e–h). Same as in (a–d) for V1 neurons. (i–l) Same as in (a–d) for neurons in higher visual areas.

The online version of this article includes the following figure supplement(s) for figure 4:

**Figure supplement 1.** Comparison of neuronal activity in virtual reality (VR) and replay conditions.

Many V1 neurons showed weaker modulation by spatial position during replay than in VR, particularly during stationary replay (*Figure 4e–h*). In V1, 'running replay' reduced median SMI by ~10%, from 0.42 ± 0.15 in VR to 0.38 ± 0.14 (n = 32 sessions, p=0.01, right-tailed Wilcoxon signed rank test, *Figure 4e,f*). This decrease in spatial modulation was not associated with mean activity differences (p>0.05, paired-sample right-tailed t-test, *Figure 4—figure supplement 1a*). Therefore, running without matched visual feedback did not result in the same spatial modulation as active navigation. The decrease in spatial modulation was greater during rest ('stationary replay', *Figure 4g,h*). As expected in mice that are not running (*Keller et al., 2012*; *Saleem et al., 2013*), activity decreased markedly (p=$10^{-24}$, paired-sample right-tailed t-test, *Figure 4—figure supplement 1b*). In addition, the median SMI halved from 0.38 ± 0.21 in VR to 0.18 ± 0.12 in stationary replay, a significant decrease (n = 24 sessions, p=0.005, right-tailed Wilcoxon signed rank test).

These effects were especially marked in HVAs (*Figure 4i–l*). Here, 'running replay' reduced median SMI by ~33%, from 0.40 ± 0.10 in VR to 0.27 ± 0.10 (n = 41 sessions, p=$10^{-6}$, right-tailed Wilcoxon signed rank test, *Figure 4i,j*), without affecting overall activity (p>0.05 in all areas, paired-sample right-tailed t-test, *Figure 4—figure supplement 1a*). Even stronger effects were seen during

stationary replay (*Figure 4k,l*), where the median SMI decreased ~68%, from 0.34 ± 0.15 in VR to 0.11 ± 0.11 (n = 33 sessions, p=$10^{-6}$, right-tailed Wilcoxon signed rank test, *Figure 4k,l*). This effect was accompanied by decreased firing in some areas, notably AM (p=0.03) and PM (p=$10^{-9}$, paired-sample right-tailed t-test, *Figure 4—figure supplement 1b*).

## Discussion

Taken together, these results indicate that upon experience active navigation modulates the amplitude of visual responses along the visual pathway and does so primarily in the cortex.

This spatial modulation of V1 visual responses strengthened across the first days of experience, perhaps more slowly than the development of navigational signals in hippocampus (*Frank et al., 2004*; *Karlsson and Frank, 2008*) but similar to retrosplenial cortex (*Mao et al., 2018*). Similar results have been observed when decoding spatial position from V1 across days of exposure to a slightly changing environment (*Fiser et al., 2016*).

The spatial modulation of V1 responses is unlikely to be inherited from LGN, because this modulation was negligible in LGN inputs to layer 4 and in LGN neurons themselves (regardless of what layer they project to *Cruz-Martín et al., 2014*). However, our mice were head restrained and hence lacked vestibular inputs, which may be relevant (*Ravassard et al., 2013*; *Russell et al., 2006*). Perhaps when mice freely move, LGN does show some spatial modulation (*Hok et al., 2018*), which is possibly amplified in V1 by nonlinear mechanisms (*Chariker et al., 2016*; *Lien and Scanziani, 2013*).

Spatial modulation affected all cortical visual areas approximately equally, consistent with the widespread neural coding of task-related information across the posterior cortex (*Koay et al., 2021*; *Minderer et al., 2019*). In addition, all areas gave stronger visual responses during active behavior than during passive viewing.

Navigational signals may reach visual cortex through retrosplenial cortex (*Makino and Komiyama, 2015*), an area that contains experience-dependent spatial signals (*Mao et al., 2017*; *Mao et al., 2018*), and is more strongly modulated by active navigation than V1 (*Fischer et al., 2020*). Another candidate is anterior cingulate cortex (*Zhang et al., 2014*), whose dense projections to V1 carry signals related to locomotion (*Leinweber et al., 2017*). The route that navigational signals take across the cortex is yet to be charted.

## Materials and methods

### Key resources table

| Reagent type (species) or resource | Designation | Source or reference | Identifiers | Additional information |
|---|---|---|---|---|
| Strain, strain background (*Mus musculus*) | WT, C57BL/6J | Jackson Labs | RRID:IMSR_JAX:000664 | |
| Strain, strain background (*Mus musculus*) | Ai93, C57BL/6J | Jackson Labs; *Madisen et al., 2015* | B6;129S6-Igs7$^{tm93.1(tetO-GCaMP6f)Hze}$/J RRID:IMSR_JAX:024103 | |
| Strain, strain background (*Mus musculus*) | Emx1-Cre, C57BL/6J | Jackson Labs; *Madisen et al., 2015* | B6.129S2-Emx1$^{(tm1(cre))Krj}$/J RRID:IMSR_JAX:005628 | |
| Strain, strain background (*Mus musculus*) | Camk2a-tTA, C57BL/6J | Jackson Labs | B6.Cg-Tg(Camk2a-tTA) 1Mmay/DboJ RRID:IMSR_JAX:007004 | |
| Strain, strain background (*Mus musculus*) | tetO-G6s, C57BL/6J | Jackson Labs; *Wekselblatt et al., 2016* | B6;DBA-Tg(tetO-GCaMP6s) 2Niell/J RRID:IMSR_JAX:024742 | |
| Recombinant DNA reagent | AAV9.CamkII. GCamp6f.WPRE.SV40 | Addgene | Catalogue #: 100834-AAV9 | |
| Software, algorithm | Suite2p | *Pachitariu et al., 2016a*; https://github.com/cortex-lab/Suite2P | RRID:SCR_016434 | |
| Software, algorithm | KiloSort | *Pachitariu et al., 2016b*; https://github.com/cortex-lab/Kilosort | RRID:SCR_016422 | |

All experimental procedures were conducted under personal and project licenses issued by the Home Office, in accordance with the UK Animals (Scientific Procedures) Act 1986.

For calcium imaging experiments in visual cortex, we used double or triple transgenic mice expressing GCaMP6 in excitatory neurons (five females, one male, implanted at 4–10 weeks). The triple transgenics expressed GCaMP6 fast (*Madisen et al., 2015*) (Emx1- Cre;Camk2a-tTA;Ai93, three mice). The double transgenic expressed GCaMP6 slow (*Wekselblatt et al., 2016*) (Camk2a-tTA; tetO-G6s, three mice). Because Ai93 mice may exhibit aberrant cortical activity (*Steinmetz et al., 2017*), we used the GCamp6 slow mice to validate the results obtained from the GCaMP6 fast mice. For calcium imaging experiments of LGN boutons, we used three C57BL/6 mice (three females, implanted at 6–9 weeks).

## Surgical procedures

To image activity in visual cortex, 4–10-week-old mice were implanted with an 8 mm circular chamber and a 4 mm craniotomy was performed over the left or right visual cortex as previously described (*Saleem et al., 2018*). The craniotomy was performed by repeatedly rotating a biopsy punch and it was shielded with a double coverslip (4 mm inner diameter; 5 mm outer diameter).

To image activity of LGN boutons, after the craniotomy was performed over the right hemisphere, we injected 253 nL (2.3 nL pulses separated by 5 s, 110 pulses) of virus AAV9.CamkII. GCamp6f.WPRE.SV40 ($5.0 \times 10^{12}$ GC/mL) into the right visual thalamus. To target LGN the pipette was directed at 2.6 mm below the brain surface, 2.3 mm posterior, and 2.25 mm lateral from bregma. To prevent backflow, the pipette was kept in place for 5 min after the end of the injection. In addition to dorsal LGN, the virus could infect neighboring thalamic nuclei, including the higher-order visual thalamic nucleus LP, which projects to layer 1 of visual cortex (*Roth et al., 2016*). Therefore, we imaged boutons only in layer 4, the main recipient of dorsal LGN inputs.

## Widefield calcium imaging

To obtain retinotopic maps we used widefield calcium imaging, as previously described (*Saleem et al., 2018*). Briefly, we used a standard epi-illumination imaging system (*Carandini et al., 2015*; *Ratzlaff and Grinvald, 1991*) together with an SCMOS camera (pco.edge, PCO AG). A 14°-wide vertical window containing a vertical grating (spatial frequency 0.15 cycles per degree), swept (*Kalatsky and Stryker, 2003*; *Yang et al., 2007*) across 135° of azimuth angle (horizontal position), at a frequency of 2 Hz. To obtain maps for preferred azimuth we combined responses to the two stimuli moving in opposite direction (*Kalatsky and Stryker, 2003*).

## Two-photon imaging

Two-photon imaging was performed with a standard multiphoton imaging system (Bergamo II; Thorlabs Inc) controlled by ScanImage4 (*Pologruto et al., 2003*). A 970 nm or 920 nm laser beam, emitted by a Ti:Sapphire Laser (Chameleon Vision, Coherent), was targeted onto L2/3 neurons or L4 LGN boutons through a 16× water-immersion objective (0.8 NA, Nikon). Fluorescence signal was transmitted by a dichroic beam splitter and amplified by photomultiplier tubes (GaAsP, Hamamatsu). The emission light path between the focal plane and the objective was shielded with a custom-made plastic cone to prevent contamination from the monitors' light. Multiple-plane imaging was enabled by a piezo focusing device (P-725.4CA PIFOC, Physik Instrumente) and an electro-optical modulator (M350-80LA, Conoptics Inc) which allowed adjustment of the laser power with depth.

For experiments monitoring activity in visual cortex, we imaged four planes in layer 2/3 set apart by 40 μm. Images of 512 × 512 pixels, corresponding to a field of view of 500 × 500 μm, were acquired at a frame rate of 30 Hz (7.5 Hz per plane). For experiments monitoring activity of LGN boutons, we imaged 7–10 planes set apart by 1.8 μm at a depth of at least 270 μm (two to three of these planes were fly-back). Images of 256 × 256 pixels, corresponding to a field of view of 100 × 100 μm, were acquired at a frame rate of 58.8 Hz.

For experiments in naïve mice (*Figure 1e–h*) we targeted the same field of view based on vasculature and recorded from similar depths. We did not attempt to track the same neurons across days.

## Neuropil receptive fields

To obtain neuropil receptive fields, on each two-photon imaging session we presented sparse uncorrelated noise for 5 min. The screen was divided into a grid of squares $4 \times 4°$. Each square was turned on and off randomly at a 10 Hz rate. At each moment, 2% of the squares were on. To compute the neuropil receptive fields, the two-photon field of view was segmented into $5 \times 5$ patches (100 μm x 100 μm surface per patch). For each patch, we averaged the fluorescence across the pixels and computed its stimulus-triggered average. The response was further smoothed in space and its peak was defined as the patch's receptive field center.

## Virtual Reality

Mice were head-restrained in the center of three LCD monitors (Ilyama ProLite E1980SD 19″) or three 10-inch LCD screens (LP097Q $\times$ 1 SPAV 9.7″, XiongYi Technology Co., Ltd.) placed at 90° angle to each other. The distance from each screen was 19 cm for the LCD monitors, or 11 cm for the LCD screen, so that visual scenes covered the visual field by 135° in azimuth and 42° in elevation.

The VR environment was a corridor with two visually matching segments (*Saleem et al., 2018*). Briefly, the corridor was 8 cm wide and 100 cm long. A vertical grating or a plaid, 8 cm wide each, alternated in the sequence grating-plaid-grating-plaid at 20, 40, 60, and 80 cm from the start of the corridor.

In VR mode, animals traversed the virtual environment by walking on a polystyrene cylindrical wheel (15 cm wide, 18 cm diameter) which allowed movement along a single dimension (forward or backward). Running speed was measured online with a rotary encoder (2400 pulses/rotation, Kübler, Germany) and was used to update the visual scene. Upon reaching the 100th cm of the corridor, mice were presented with a gray screen for an inter-trial period of 3–5 s (chosen randomly), after which they were placed at the beginning of the virtual corridor for the next trial. The duration of each trial depended on how long it took the mouse to traverse the corridor, typically <8 s. Trials in which animals did not reach the end of the corridor within 30 s were timed-out and excluded from further analysis. A typical session included >50 trials.

In the replay mode, mice were presented with a previous closed-loop session, while still free to run on the wheel.

## Electrophysiology

Mice were implanted with a custom-built stainless-steel metal plate on the skull under isoflurane anesthesia. The area above the right LGN was kept accessible for electrophysiological recordings. Mice were acclimatized to the VR environment for >5 days. The virtual corridor was projected onto a truncated spherical screen, and the mice traversed it by running on a 10 cm radius polystyrene ball (*Schmidt-Hieber and Häusser, 2013*). Twelve to twenty-four hours before the first recording, a ~1 mm craniotomy was performed over the LGN (1.9 mm lateral and 2.4 mm anterior from lambda). On the recording session, a multi-shank electrode (ASSY-37 E-1, 32-channels, Cambridge Neurotech Ltd., Cambridge, UK) was advanced to a depth of ~3 mm until visual responses to flashing stimuli were observed. Electrophysiology data were acquired with an OpenEphys acquisition board (*Siegle et al., 2017*) and units were isolated using Kilosort (*Pachitariu et al., 2016b*).

## Behavior and training

Mice ran through the corridor with no specific task (n = 4 animals, 65 sessions recording cortical activity; n = 3 animals, 19 sessions recording activity of LGN boutons). Prior to recording sessions, mice were placed in the virtual environment, typically for 3 days and for up to 1 week, until they were able to run for at least 80% of the time within a single session. For our experiments in cortex, mice ran most of the time without rewards (34/65 sessions). If mice became slower in subsequent sessions, they were motivated to run with rewards, receiving ~2.5 μL of water (two mice) or of 10% sucrose (one mouse) through a solenoid valve (161T010; Neptune Research, USA). In 12/65 sessions, rewards were placed at random positions along the corridor. In 19/65 sessions rewards were placed at the end of the corridor. We chose various reward protocols to control for the possible effect of reward and of stereotyped running speeds that might be observed with rewards at the end of the corridor. Our results were the same regardless of whether animals received rewards or not

(*Figure 3—figure supplement 3*). Therefore, rewards were placed at the end of the virtual corridor for subsequent recordings from LGN boutons.

For experiments in naïve animals (n = 2, Camk2a-tTA;tetO-G6s, *Figure 1e–h*) mice were placed on the treadmill for 4–5 days, until they were able to run at speeds higher than 10 cm/s for at least 20 min. Only after animals reached this criterion we turned the VR on and started recording from the same field of view in V1 across multiple days.

We tracked the eye of the animal using an infrared camera (DMK 21BU04.H, Imaging Source) and custom software, as previously described (*Saleem et al., 2018*).

## Perfusion and histology

Mice were perfused with 4% paraformaldehyde (PFA) and the brain was extracted and fixed for 24 hr at 4°C in 4% PFA, then transferred to 30% sucrose in PBS at 4°C. The brain was mounted on a benchtop microtome and sectioned at 60 µm slice thickness. Free-floating sections were washed in PBS, mounted on glass adhesion slides, stained with DAPI (Vector Laboratories, H-1500), and covered with a glass-slip. In brains used for two-photon imaging we obtained anatomical images in blue for DAPI and in green for GCaMP. In brains used for electrophysiology we obtained anatomical images in blue for DAPI and red for DiI (the electrode had been dipped in DiI before insertion). The LGN border on these images was determined using SHARP-Track (*Shamash et al., 2018*).

## Processing of two-photon imaging data

Image registration in the horizontal plane (x–y), cell detection, and spike deconvolution were performed with Suite2p (*Pachitariu et al., 2016a*; *Pachitariu et al., 2018*). All subsequent analyses were performed on each neuron's activity inferred from spike deconvolution. To account for the different dynamics of the calcium indicators, the decay timescale used for deconvolution was set to 0.5 s for GCaMP6f and to 2.0 s for GCaMP6s.

For the LGN boutons data, we additionally used the method of *Schröder et al., 2020* to align image frames in the z-direction (cortical depth). By using a stack of closely spaced planes (1.8 µm inter-plane distance), we were able to detect small boutons across multiple planes, which could have otherwise moved outside a given plane due to brain movement in the z-direction. In brief, for each imaging stack, the algorithm estimates the optimal shift that maximizes the similarity of each plane to their corresponding target image (with target images across planes having been aligned to each other in x and y). After assigning the shifted planes to their corresponding target image, a moving average across two to three neighboring planes was applied, resulting in a smooth image, and consequently in smooth calcium traces from boutons sampled from multiple, closely spaced planes.

Regions of interest (cell bodies or boutons) were detected from the aligned frames and were manually curated with the Suite2p graphical user interface, as described by *Saleem et al., 2018*. Data from V1 neurons with receptive fields in the periphery (>40°) are the same as in *Saleem et al., 2018*. These data were deconvolved and pooled together with data from V1 neurons with receptive fields in the central visual field.

## Analysis of responses in VR

To obtain response profiles as a function of position along the corridor, we first smoothed the deconvolved traces in time with a 250 ms Gaussian window and considered only time points with running speeds greater than 1 cm/s. We then discretized the position of the animal in 1 cm bins (100 bins) and estimated each neuron's spike count and the occupancy map, that is the total time spent in each bin. Both maps were smoothed in space with a fixed Gaussian window of 5 cm. Finally, each unit's response profile was defined as the ratio of the smoothed spike count map over the smoothed occupancy map. We assessed the reliability of the response profiles based on a measure of variance explained and selected those with variance explained higher than 5%.

To predict the responses that would be observed if cells were purely visual (dotted curves in *Figure 1b*), we fit (using least squares) a smooth function to the response profile along the visually matching segment where the cell peaked. The smooth function was the sum of two Gaussians that meet at the peak. To obtain a prediction along the whole corridor, we then duplicated the fitted response at ±40 cm away from the maximum.

To cross-validate the response profile patterns in VR, we divided each session's trials in two halves (odd vs. even) and obtained a pair of response profiles for each unit. Odd trials were used as the train set to determine the position at which cells or boutons preferred to fire maximally. Odd trials were subsequently excluded from further analysis.

The same splitting into odd and even trials was used to estimate each unit's SMI. For each neuron or bouton, the position of the peak response was measured from the response profile averaged across odd trials ('preferred position'). We then obtained the response, $R_{p,}$, at the preferred position and the visually identical position 40 cm away ('non-preferred position': $R_n$), using the response profile averaged across even trials. Units with maximal response close to the start or end of the corridor (0–15 cm or 85–100 cm) were excluded, because their preferred position fell outside the visually matching segments. SMI was defined as:

$$\text{SMI} = \frac{R_p - R_n}{R_p + R_n}$$

Therefore, a response with two equal peaks would have $\text{SMI} = 0$, whereas a response with one peak would have $\text{SMI} = 1$. SMI would be negative if there was no consistent preference, that is, the larger response was in one half of the corridor in odd trials and in the other half in even trials.

To cross-validate the response profile patterns and to estimate SMIs in passive viewing, we used the same odd trials from VR as a train set. Based on those we obtained response profile patterns and SMIs from all trials during passive viewing. To isolate periods when the animal was stationary during passive viewing, we considered only times when the speed of the animal was <5 cm/s. Response profiles during stationary viewing were estimated only in sessions where the animal was stationary in at least 10 trials. To isolate periods when the animal was running during passive viewing, we considered only times when the speed of the animal was >1 cm/s. Response profiles in running during replay were estimated only in sessions where the animal was running in at least 10 trials.

The reliability of a neuron's or bouton's activity was defined as the variance in activity explained by the cross-validated response profile. To predict activity, data were divided into fivefolds (fivefold cross-validation) and activity for each fold was predicted from the responses profile estimated from all other folds (training data). Reliability was defined as:

$$\text{Reliability} = 1 - \frac{\sum_t \left(y(t) - y^{'}(t)\right)^2}{\sum_t \left(y(t) - \mu\right)^2} \tag{1}$$

where $y(t)$ is the actual, smoothed activity at time $t$ between the beginning and end of the experiment, $y^{'}(t)$ is the predicted activity for the same time bin, and $\mu$ is the mean activity of the training data. The response reliability reported was obtained from the mean reliability across folds. Only neurons or boutons with response reliability > 5% were considered for further analysis.

## General linear models

To assess the joint contribution of all visual and behavioral factors in VR we fitted the V1 deconvolved responses to three multilinear regression models similar to *Saleem et al., 2018*. The models had the form: $y = X\beta$, where $X$ is a T-by-M matrix with $T$ time points and $M$ predictors and $y$ is the predicted calcium trace (T-by-1 array). Optimal coefficient estimates $\beta$ (M-by-1 array) that minimize the sum-squared error were obtained using:

$$\hat{\beta} = \left(X^T X + \lambda I\right)^{-1} X^T y,$$

where $\lambda$ is the ridge-regression coefficient.

The simplest model, the *visual* model, relied only on 'trial onset' (first 10 cm in the corridor), 'trial offset' (last 10 cm in the corridor), and the repetition of visual scenes within the visually matching segments (from 10 to 90 cm in the corridor). The basis functions for all predictors were square functions with width of 2 cm and height equal to unity. To model the repetition of visual scenes, a predictor within the visually matching segments comprised of two square functions placed 40 cm apart. Thus, the visual model's design matrix had 30 predictors plus a constant: five predictors for trial onset, five predictors for trial offset, and 20 predictors within the visually matching segments.

The second model, the *non-spatial* model, extended the *visual* model to assess the influence of all the behavioral factors we measured: running speed, reward events, pupil size, and the horizontal and vertical pupil position. These factors were added as predictors to the design matrix of the *visual* model, as follows: running speed was shifted backward and forward in time twice, in 500 ms steps, thus contributing five continuous predictors; pupil size and horizontal and vertical pupil position contributed one continuous predictor each; each reward event contributed one binary predictor at the time of the reward. The continuous predictors of running speed and pupil size were normalized between 0 and 1, whereas pupil position was normalized between $-1$ and 1 to account for movements in opposite directions.

The third model, the *spatial* model, extended the *non-spatial* model by allowing for an independent scaling of the two visually matching segments in the corridor. For each predictor within the visually matching segments, the two square functions were allowed to have different heights. The height of the two square functions was parameterized by a parameter $\alpha$, such that the two functions had unit norm. An $\alpha = 0.5$ corresponded to a purely visual representation with SMI $\sim 0$, while $\alpha = 1$ or $\alpha = 0$ would correspond to a response only in the first or second segment, and an SMI $\sim 1$.

To choose the best model, we used the ridge regression coefficient, $\lambda$, that maximized the variance explained using fivefold cross-validation, searching the values $\lambda = 0.01, 0.05, 0.1, 0.5,$ or 1. In the spatial model, we performed multiple ridge regression fits, searching for the optimal value of $\alpha$ using a step size of 0.1, for each $\lambda$.

The predictions obtained were subsequently processed similarly as the original deconvolved traces to obtain predicted response profiles and SMI.

## Acknowledgements

We thank Julien Fournier for helpful discussions, Michael Krumin for assistance with two-photon imaging, and Karolina Socha for advice on imaging LGN boutons. This work was supported by EPSRC (PhD award F500351/1351 to EMD), by a Wellcome Trust/Royal Society Sir Henry Dale Fellowship (200501 to ABS), by a Human Frontiers in Science Program (grant RGY0076/2018 to ABS), and by the Wellcome Trust (grant 205093 to MC and KDH). MC holds the GlaxoSmithKline/Fight for Sight Chair in Visual Neuroscience.

## Additional information

### Funding

| Funder | Grant reference number | Author |
|---|---|---|
| EPSRC | F500351/1351 | E Mika Diamanti |
| Wellcome Trust/ Royal Society | Sir Henry Dale fellowship 200501 | Aman B Saleem |
| Wellcome Trust | 205093 | Kenneth D Harris Matteo Carandini |
| Human Frontier Science Program | RGY0076/2018 | Aman B Saleem |

The funders had no role in study design, data collection and interpretation, or the decision to submit the work for publication.

### Author contributions

E Mika Diamanti, Conceptualization, Software, Formal analysis, Funding acquisition, Investigation, Visualization, Methodology, Writing - original draft, Writing - review and editing; Charu Bai Reddy, Investigation, Methodology; Sylvia Schröder, Software, Methodology, Writing - review and editing; Tomaso Muzzu, Investigation; Kenneth D Harris, Conceptualization, Resources, Supervision, Funding acquisition, Writing - review and editing; Aman B Saleem, Conceptualization, Resources, Software, Supervision, Funding acquisition, Methodology, Writing - original draft, Writing - review and editing; Matteo Carandini, Conceptualization, Resources, Supervision, Funding acquisition, Methodology, Writing - original draft, Writing - review and editing

## Author ORCIDs

E Mika Diamanti [ID] https://orcid.org/0000-0003-1199-3362
Charu Bai Reddy [ID] https://orcid.org/0000-0002-8195-3326
Sylvia Schröder [ID] http://orcid.org/0000-0002-9938-3931
Tomaso Muzzu [ID] https://orcid.org/0000-0002-0018-8416
Kenneth D Harris [ID] http://orcid.org/0000-0002-5930-6456
Aman B Saleem [ID] https://orcid.org/0000-0002-7100-1954
Matteo Carandini [ID] http://orcid.org/0000-0003-4880-7682

## Ethics

Animal experimentation: All experimental procedures were conducted under personal and project licenses issued by the Home Office, in accordance with the UK Animals (Scientific Procedures) Act 1986.

## Decision letter and Author response

Decision letter https://doi.org/10.7554/eLife.63705.sa1
Author response https://doi.org/10.7554/eLife.63705.sa2

## Additional files

### Supplementary files

• Transparent reporting form

### Data availability

Data presented in the main figures of this study are uploaded in Dryad. In addition, we have uploaded the full imaging dataset acquired in this study. The dataset includes deconvolved traces of all imaged cells (or boutons) and all relevant behavioral variables (https://doi.org/10.5061/dryad.4j0zpc893).

The following dataset was generated:

| Author(s) | Year | Dataset title | Dataset URL | Database and Identifier |
|---|---|---|---|---|
| Diamanti EM, Reddy CB, Schröder S, Muzzu T, Harris KD, Saleem AB, Carandini M | 2021 | Spatial modulation of visual responses arises in cortex with active navigation; main figures and full dataset | http://dx.doi.org/10.5061/dryad.4j0zpc893 | Dryad Digital Repository, 10.5061/dryad.4j0zpc893 |

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
