## [Decision Letter]

**Acceptance summary:**

This paper investigates the modulation of spatial signals in higher order visual areas in mice navigating virtual reality environments. While previous work demonstrated that the spatial position of an animal modulates neural activity in primary visual cortex (V1), the authors demonstrate that this spatial modulation is not a general feature of the visual circuit. Rather, spatial modulation of the visual circuit only occurs in higher visual areas, not in lower visual areas, and reflects the animal's experience in a particular environment and active engagement in navigation. Together, this work reveals that spatial modulation of neurons in the visual system is likely generated within the visual cortex itself, rather than inherited in a bottom-up manner from the thalamus.

**Decision letter after peer review:**

Thank you for submitting your article "Spatial modulation of visual signals arises in cortex with active navigation" for consideration by *eLife*. Your article has been reviewed by two peer reviewers, and the evaluation has been overseen by a Reviewing Editor and Laura Colgin as the Senior Editor. The reviewers have opted to remain anonymous.

The reviewers have discussed the reviews with one another and the Reviewing Editor has drafted this decision to help you prepare a revised submission.

Summary:

This paper investigates the modulation of spatial signals in higher order visual areas in mice navigating virtual reality environments. Previous work demonstrated that the spatial position of an animal modulates neural activity in primary visual cortex (V1). Here, the authors demonstrate that this spatial modulation however, is not a general feature of the visual circuit. Similar spatial modulation occurs in higher visual areas but not in lower visual areas, such as the lateral geniculate nucleus. Moreover, this work finds that spatial modulation was stronger when animals had more experience on the track and when the animals were actively performing a task, rather than when the animal was passively viewing the same virtual track. Since the first reports that visual neurons show modulation by spatial position during spatial navigation tasks, similar to that observed in hippocampal place cells, the source of this modulation has been an open question. This work adds new insight regarding this question, suggested that it is likely either generated within the visual cortex itself of propagated in a top-down manner from higher brain areas, rather than in a bottom-up manner from the thalamus.

Revisions:

1) The imaging data is from mice with different genetic backgrounds, as well as a mixture of gcamp6f and 6s. In addition, different reward protocols were used for different mice. Although the authors state in the Materials and methods that none of these factors impact their results, please include some quantifications to support this statement (e.g. the distribution of SMI for 6f data vs 6s data).

These differences could affect portions of the results where the dataset is split up – for example in the comparison between different higher visual areas, and the observation that spatial modulation appears to vary with receptive field location.

2) The authors state that it is to be expected that LGN neurons respond more strongly in the first half of the corridor due to contrast adaption mechanisms. Please provide a quantification that supports this statement.

3) For the spatial modulation index, it appears that the reported values switch between median (e.g. Figure 1 and 2) and mean (Figure 4), t-test and rank-sum – and sometimes there is missing information regarding which (mean or median) is being reported. Please include more detail regarding when/why each value was reported and which value is reported.

4) It was not clear if imaging was only performed on layer 2/3 neurons or if it includes deeper layers.

5) Throughout the paper, the authors use “firing rate” to refer to deconvolved calcium signal. Although this is stated in the Materials and methods, this wording may be confusing, especially since the paper also contains extracellular recordings of spiking activity.

6) It was not clear how the dotted lines (e.g. Figure 1B) were calculated.

7) Because experience and task engagement enhanced spatial modulation, the authors concluded in the Abstract that "Active navigation in a familiar environment, therefore, determines spatial modulation…". However, this conclusion is too strong and not well-supported by the data. First, spatial modulation on Day 1, when the task was novel, was lower than on later days, but it was already much higher than 0 (Figure 1H). In addition, the individual neuron data (Figure 1E) displays clear spatial modulation on Day 1. Therefore, "familiar environment" is not a requirement. Second, spatial modulation during passive viewing was much higher than 0 and was correlated with that during active navigation, as shown in Figure 4E – Figure 4L. Therefore, "active navigation" is not a requirement either. It is true that both active navigation and familiar environment enhanced spatial modulation. They did not however, "determine" spatial modulation.

8) Related to the point above, the presence of spatial modulation in passive viewing indicates that these cells in the visual system were still mainly driven by visual stimuli. The data in Figure 4E,F are especially telling: the modulation in V1 was similar and highly correlated between active navigation and running replay. In addition, it is clear from all the raw traces in Figure 1 and Figure 2 that these cells did respond to the two segments with identical stimuli reliably with two peaks. The spatial modulation was just a change in one of the peaks. So the nature of the modulation is a "rate remapping" of the expected, classical visual responses. In order to maintain the big picture of what drives the activity of these neurons, it would be beneficial to clarify that the "spatial modulation" is a modulation on top of the expected visual responses. This message is not explicitly conveyed in the current manuscript.

9) The authors stated that spatial modulation is "largely absent in the main thalamic pathway into V1". This was based on the significantly weaker SMIs in LGN than those in V1 and HVAs. However, it is unclear whether the SMIs in LGN were still significant. The SMI values for both LGN buttons and LGN units might be statistically significant from zero. The statistical comparison p-values should be given in both cases. Second, Figure 3—figure supplement 1B,F show that the SMI values in LGN could be predicted by spatial modulation, but not by visual stimuli alone or behavioral variations, just like those in V1 and HVAs. This seems to be good evidence for the presence of spatial modulation in LGN. Therefore, it seems the data do not support the complete lack of spatial modulation in LGN, but do clearly demonstrate weaker spatial modulation in LGN than in V1 and HVAs.

---

## [Author Response]

Revisions:1) The imaging data is from mice with different genetic backgrounds, as well as a mixture of gcamp6f and 6s. In addition, different reward protocols were used for different mice. Although the authors state in the Materials and methods that none of these factors impact their results, please include some quantifications to support this statement (e.g. the distribution of SMI for 6f data vs 6s data).These differences could affect portions of the results where the dataset is split up – for example in the comparison between different higher visual areas, and the observation that spatial modulation appears to vary with receptive field location.

We see the reviewer’s point and we have added a new supplementary figure (Figure 3—figure supplement 3) to demonstrate that the distribution of spatial modulation indices (SMIs) does not depend on mouse line or reward protocol. We estimated the SMI separately for different mouse lines and reward protocols. Because we did not image each combination of visual area and mouse line or reward condition, we pooled the higher visual areas in a single plot. We thus show the results for overall V1, for V1 center and periphery, and for the higher visual areas. We refer to these controls in Results. We now also report the number of sessions per condition in Materials and methods.

2) The authors state that it is to be expected that LGN neurons respond more strongly in the first half of the corridor due to contrast adaption mechanisms. Please provide a quantification that supports this statement.

Contrast adaptation would cause the responses to be larger in the first half than in the second half. This is a qualitative statement, and it would be hard to turn it into a credible quantitative statement, because previous studies of adaptation in LGN have used very different stimuli. To help clarify this we have replaced “would” with “might”, to read “as might be expected from contrast adaptation mechanisms”. We have also strengthened the quantification of the results: we included the exact number of neurons with SMI > 0.5 and preferred position in the first visually-matching segment (28/37).

3) For the spatial modulation index, it appears that the reported values switch between median (e.g. Figure 1 and 2) and mean (Figure 4), t-test and rank-sum – and sometimes there is missing information regarding which (mean or median) is being reported. Please include more detail regarding when/why each value was reported and which value is reported.

We thank the reviewers for spotting this. For consistency, we now test for the median SMI throughout, using Wilcoxon tests (rank sum or signed rank). All values we report are now median ± m.a.d. We also note that whenever possible we perform statistical tests on the median SMI across sessions rather than neurons, because neuronal responses recorded simultaneously might not be independent.

4) It was not clear if imaging was only performed on layer 2/3 neurons or if it includes deeper layers.

Thank you for spotting this. We now explain that imaging of neurons was performed in layer 2/3 in Results and in Materials and methods.

5) Throughout the paper, the authors use “firing rate” to refer to deconvolved calcium signal. Although this is stated in the Materials and methods, this wording may be confusing, especially since the paper also contains extracellular recordings of spiking activity.

We agree and we have now reserved the term “firing rate” for the electrophysiological measurements. For the optical measurements we now use the term “activity” and specify that “activity” stands for deconvolved Calcium traces.

6) It was not clear how the dotted lines (e.g. Figure 1B) were calculated.

Thank you for spotting this. We added this information in Materials and methods.

7) Because experience and task engagement enhanced spatial modulation, the authors concluded in the Abstract that "Active navigation in a familiar environment, therefore, determines spatial modulation…". However, this conclusion is too strong and not well-supported by the data. First, spatial modulation on Day 1, when the task was novel, was lower than on later days, but it was already much higher than 0 (Figure 1H). In addition, the individual neuron data (Figure 1E) displays clear spatial modulation on Day 1. Therefore, "familiar environment" is not a requirement. Second, spatial modulation during passive viewing was much higher than 0 and was correlated with that during active navigation, as shown in Figure 4E – Figure 4L. Therefore, "active navigation" is not a requirement either. It is true that both active navigation and familiar environment enhanced spatial modulation. They did not however, "determine" spatial modulation.

We agree, and we have corrected the Abstract to reflect this. We now say that modulation “strengthens with” (rather than “requires engagement in”) active behavior, and that active navigation in a familiar environment “enhances” (rather than “determines”) the spatial modulation.

8) Related to the point above, the presence of spatial modulation in passive viewing indicates that these cells in the visual system were still mainly driven by visual stimuli. The data in Figure 4E,F are especially telling: the modulation in V1 was similar and highly correlated between active navigation and running replay. In addition, it is clear from all the raw traces in Figure 1 and Figure 2 that these cells did respond to the two segments with identical stimuli reliably with two peaks. The spatial modulation was just a change in one of the peaks. So the nature of the modulation is a "rate remapping" of the expected, classical visual responses. In order to maintain the big picture of what drives the activity of these neurons, it would be beneficial to clarify that the "spatial modulation" is a modulation on top of the expected visual responses. This message is not explicitly conveyed in the current manuscript.

We agree completely, and this is exactly how we think of the effect: it’s a modulation of a visual response. This is why we call it “spatial modulation”. To make sure that this is clear, we now have reworded the Results and multiple sentences in the Discussion, including the first one. We also substituted the terms “responses” or “visual signals” with “visual responses” across the manuscript and in the title.

9) The authors stated that spatial modulation is "largely absent in the main thalamic pathway into V1". This was based on the significantly weaker SMIs in LGN than those in V1 and HVAs. However, it is unclear whether the SMIs in LGN were still significant. The SMI values for both LGN buttons and LGN units might be statistically significant from zero. The statistical comparison p-values should be given in both cases. Second, Figure 3—figure supplement 1B,F show that the SMI values in LGN could be predicted by spatial modulation, but not by visual stimuli alone or behavioral variations, just like those in V1 and HVAs. This seems to be good evidence for the presence of spatial modulation in LGN. Therefore, it seems the data do not support the complete lack of spatial modulation in LGN, but do clearly demonstrate weaker spatial modulation in LGN than in V1 and HVAs.

Thank you for this suggestion. We have now added the p-value for statistical significance for the SMI of LGN boutons and neurons. As the reviewer suspected, they are small but significantly different from zero. Crucially, they are markedly lower than those measured in cortical visual areas. We modified the text accordingly in Abstract (“negligible” instead of “largely absent”), Results and Discussion.